# G-Protein Subunit Gamma 4 as a Potential Biomarker for Predicting the Response of Chemotherapy and Immunotherapy in Bladder Cancer

**DOI:** 10.3390/genes13040693

**Published:** 2022-04-14

**Authors:** Lianhui Duan, Xuefei Liu, Ziwei Luo, Chen Zhang, Chun Wu, Weiping Mu, Zhixiang Zuo, Xiaoqing Pei, Tian Shao

**Affiliations:** 1School of Life Sciences, Zhengzhou University, Zhengzhou 450001, China; 202022162012918@gs.zzu.edu.cn (L.D.); 202012162012893@gs.zzu.edu.cn (C.Z.); 2State Key Laboratory of Oncology in Southern China, Collaborative Innovation Center for Cancer Medicine, Sun Yat-sen University Cancer Center, Guangzhou 510060, China; liuxf@sysucc.org.cn (X.L.); luozw@sysucc.org.cn (Z.L.); wuchun@sysucc.org.cn (C.W.); muweiping@rjmart.cn (W.M.); zuozhx@sysucc.org.cn (Z.Z.)

**Keywords:** bladder cancer, G-protein subunit gamma 4, immune infiltration, immunotherapy, chemotherapy

## Abstract

Background: GNG4, a member of the G-protein γ family, is a marker of poor overall survival (OS) rates in some malignancies. However, the potential role of GNG4 in bladder cancer (BLCA) is unknown. It is also unclear whether GNG4 may be utilized as a marker to guide chemotherapy or immunotherapy. Methods: Single-cell RNA sequencing data were used to explore the expression of GNG4 in tumor microenvironment of BLCA. Bulk RNA sequencing data from TCGA were used to evaluate the relationship between GNG4 expression and biological features, such as immune cell infiltrations and gene mutations. The associations between GNG4 expression and survival in BLCA patients under or not under immunotherapy were evaluated using seven BLCA cohorts. Results: GNG4 was specifically expressed in exhausted CD4^+^ T cells. And the high expression of the GNG4 was associated with high level of immune cell infiltration. The high-GNG4-expression group displayed a better response to immunotherapy, whereas patients in the low-GNG4-expression group often benefited from chemotherapy. Moreover, the high-GNG4 group was more similar to the basal group, whereas the low-GNG4 group was similar to the luminal group. Conclusions: GNG4 may be a potential biomarker for the prediction of the response to therapy in BLCA. Higher GNG4 expression can be used as a predictor of response to immunotherapy, and lower GNG4 expression can be used as a predictor of response to chemotherapy.

## 1. Introduction

Bladder cancer (BLCA), a malignancy of the urinary tract, is the second most common urinary cancer, with approximately 573,278 new cases and 212,536 deaths reported in 2020, which means it ranks as the 10th most frequently diagnosed cancer worldwide [1,2,3]. It is urgent to develop effective treatment strategies to improve the survival rate of BLCA. Chemotherapy plays an important role in the comprehensive treatment of cancer. However, tumor recurrence often occurs after chemotherapy, and over the last 30 years, there have been no significant advancements in the treatment of BLCA [4]. Despite the advent of neoadjuvant and adjuvant chemotherapy, the treatment outcome and prognosis of metastatic BLCA remain unsatisfactory [5]. Cancer immunotherapy, such as immune checkpoint blockade (ICB), has resulted in promising improvements in survival rates in individuals with advanced BLCA [6,7], which is characterized by a high tumor mutation burden (TMB) and neoantigens [8]. However, only a minority of patients respond to ICB because they are resistant to the primary or secondary mechanisms of ICB [6,7]. Although the advent of immunotherapy provided another excellent avenue for the treatment of BLCA, the results continue to be unsatisfactory [9]. Therefore, both chemotherapy and immunotherapy need to continue to improve in terms of how effectively they can treat BLCA. In general, there is an urgent need to develop novel biomarkers that can be used to effectively predict treatment response in BLCA patients.

As a member of the G-protein γ family, GNG4 normally transduces signals from upstream G-protein-coupled receptors (GPCRs) [10,11,12,13]. Incidentally, GPCRs consist of a large family of receptors that respond to a variety of extracellular stimuli such as hormones, growth factors, light, and other sensory stimulus signals [14]. Several G-protein family members have been determined to be pan-inhibitors and oncogenes of Gβγ, which reduce cell proliferation and impede metastasis in breast and prostate cancer cell lines [15,16]. Meanwhile, GNG4 has been reported to be a tumor suppressor gene in glioblastoma [14]. GNG4 is also used as a biomarker in gastric cancer, especially in liver metastasis of gastric cancer. High levels of GNG4 in primary tumor tissue have been reported to be associated with short overall survival times and the likelihood of liver metastasis recurrence [17]. Notably, GNG4 is not only a candidate biomarker for the diagnosis of synchronous metastasis and the prediction of metachronous metastasis but may also be a predictive factor concerning 5FU, a representative pyrimidine drug [17]. GNG4 was also demonstrated to be the key element of the colorectal carcinoma (CRC) tumor mutation burden (TMB), which is essential information in the ICB therapy of CRC [18]. However, the role of GNG4 in BLCA remains unclear.

In this study, we integrated the public bulk and single-cell RNA sequencing data to comprehensively investigate the GNG4 expression in tumor microenvironment and its relationship with immunotherapy response in bladder cancer.

## 2. Materials and Methods

### 2.1. Data Collection and Processing

The RNA sequencing (RNA-seq) transcriptome data of patients with BLCA and the corresponding clinical data and mutation profiles were downloaded from The Cancer Genome Atlas (TCGA) database (https://portal.gdc.cancer.gov/, accessed on 12 September 2021). Fragments per kilobase million (FPKM) values were converted to transcripts per kilobase million (TPM) values, and the Ensembl gene IDs of the RNA-seq data were converted to gene symbols concerning the annotation file.

Three BLCA validation datasets were included for analysis from the Gene Expression Omnibus (GEO) data portal (http://www.ncbi.nlm.nih.gov/geo/, accessed on 24 September 2021) (GSE13507, GSE32894, and GSE70691) [19,20,21]. The raw data were downloaded as microarray data.

We included three immunotherapeutic cohorts and obtained the relative transcriptomic and clinical data from the online Appendix A appended to the published studies. We downloaded RNA sequencing of the immunotherapy cohort of BLCA from Mvigor210CoreBiologies, an R package [22]. Another immunotherapy cohort of BLCA (GSE176307) was included for analysis from the Gene Expression Omnibus (GEO) database (http://www.ncbi.nlm.nih.gov/geo/, accessed on 24 September 2021) [23]. RNA sequencing of the immunotherapy cohort of melanoma (PRJEB23709) was downloaded from GEO [24].

The copy number variation (CNV) and somatic mutation profiles were also obtained from TCGA data. The somatic mutation data sorted in the form of Mutation Annotation Format (MAF) were analyzed using the R package ‘maftools’.

### 2.2. Single-Cell Transcriptome Sequencing and Data Preprocessing

Single-cell transcriptome sequencing data, downloaded from the Genome Sequence Archive (GSA) database (https://ngdc.cncb.ac.cn/gsa/, accessed on 15 October 2021), were analyzed using the Seurat (v3.1.3) R toolkit [25]. First, cells of poor quality were filtered out [26]. Then, DoubletFinder was used with the default parameters to predict cell doublets. Next, the “NormalizeData” function was used to normalize the data, and the “FindVariableFeatures” function was used to choose the highly variable genes. Then, the gene expression matrices of all of the samples were integrated using the “Runharmony” functions [27]. Finally, we determined the different cell types with default parameters; the different cell types were visualized using UMAP.

We used the following rules to annotate the different cell types: Based on using the “wilcoxauc” function in presto to derive the top ten differentially expressed genes, we identified 8 main clusters and annotated them according to the expression of typical gene markers, including endothelial cells, epithelial cells, fibroblast cells, myeloid cells, mast cells, plasma cells, B cells, and T cells. CD4 and CD8 gene expressions were used for the differentiation of CD4^+^ and CD8^+^ T cells. CD4^+^ T-cell subclusters were named using the first marker gene.

### 2.3. Functional and Pathway Enrichment Analysis of Bulk RNA-Seq

The identification of differentially expressed genes between the low-GNG4 and the high-GNG4 groups was carried out using the Limma R package [28]. To explore the phenotype-specific signaling pathways of the tumor microenvironment (TME) in the high-GNG4 and low-GNG4 groups, the Gene Set Enrichment Analysis (GSEA) approach was used with an adjusted *p* < 0.05 using the ‘clusterfiler’ R package [29].

### 2.4. Immunological Characteristics of the TME in BLCA

The proportions of immune cell types (i.e., TICs) evaluated for immune cell infiltration in each sample (with immune infiltration scores) were computed using the Timer [30], MCP_counter [31], CIBERSORT, and Xcell [32] algorithms.

### 2.5. Prediction of the Molecular Subtypes in BLCA

The molecular subtype of each individual was determined using the ConsensusMIBC R package [33]. All of the patients in the TCGA database were divided into six phenotypes.

### 2.6. Statistical Analysis

Using the Kaplan–Meier method to estimate OS or FPS, the Kaplan–Meier curves were compared using the log-rank test. A two-sided *p*-value of less than 0.05 was regarded as significant. Meanwhile, all of the sample sizes were large enough to enable proper statistical analysis. Spearman correlation analysis was applied in all of the correlation analyses. GraphPad Prism (https://www.graphpad.com/, accessed on 25 March 2022) was also used to perform statistical analyses. *p*-values less than 0.05 were deemed to be statistically significant. All of the t-test analyses were two-sided t-tests (paired or unpaired, depending on the experiments).

## 3. Result

### 3.1. GNG4 Is a Biomarker of Exhausted CD4^+^ T Cells in BLCA

Unsupervised clustering of the published BLCA single-cell RNA sequencing data revealed eight cell clusters (Figure 1A and Appendix A). Intriguingly, GNG4 was specifically expressed in T cells (Figure 1B, Appendix A). After dividing all CD4^+^ T cells into six sub-clusters, we found GNG4 only expressed in cluster CD4_CXCL13 (GNG4 was not expressed in CD8^+^ T cells, and the results are not shown) (Figure 1C,D and Appendix A). In addition to the high expression of GNG4, this cluster is accompanied by the high expression of PDCD1, LAG3, TIGIT, and CXCL13, suggesting that the cells of this cluster were in an exhausted state (Figure 1E). Overall, we determined that GNG4 is specifically expressed in exhausted CD4^+^ T cells in BLCA tumor microenvironment.

### 3.2. High Expression of GNG4 Reveals High Immune Infiltration but Tends to Be Exhausted

Since GNG4 was found as a biomarker to measure the exhausted CD4^+^ T cells, we next used the bulk RNA sequencing data to further explore the relationship between GNG4 and immune cell infiltration. Hence, we obtained and analyzed the clinical data of 406 patients diagnosed with BLCA in the TCGA database (Appendix A) and investigate the association between the level of GNG4 expression and immune-related indicators. The result reveals that GNG4 was positively correlated with many immunomodulators, including chemokines, receptors, MHC, and immunostimulators. (Figure 2A). Many chemokines, such as CCL3, CCL4, CCL7, CCL8, CCL11, CXCL9, CXCL10, and paired receptors, including CCR1, CCR2, CCR4, CXCR4, and CXCR6, were also positively correlated with GNG4. These chemokines and receptors promote the recruitment of effector TIICs, including TH17 cells, antigen-presenting cells, and CD8^+^ T cells [34]. Among them, two vital chemokines (CXCL9 and CXCL10), which trigger the recruitment of CD8^+^ T cells into the tumor microenvironment (TME) in BLCA, were upregulated in the high-GNG4 group. In addition, a majority of MHC molecules, especially Human Leukocyte Antigen (HLA) molecules, were highly expressed in the high-GNG4 group, which suggested the upregulation of antigen presentation and processing ability in the high-GNG4 group. Additionally, many immunostimulators also had high expression in the high-GNG4 group, which were able to stimulate the immune system by inducing activation or increasing the activity of any of its components. The same result was obtained in three BLCA validation cohorts including GSE13507, GSE32894, and GSE70691 (Appendix A). Surprisingly, in the present study, we found that GNG4 was positively related to many immune checkpoint inhibitors, including PDCD1, CTLA4, LAG3, IDO1, HAVCR2, and TIGIT (Figure 2B). This demonstrated that the high-GNG4 group had an inflammatory but inhibitory TME. Similarly, we also found a positive correlation between GNG4 and immune checkpoint inhibitors in three validation cohorts (Appendix A). We further analyzed the differential expression of genes in the high- and low-GNG4 groups in BLCA (Figure 2C, Appendix A). Based on these results, the biological functions of gene sets in high-GNG4 groups were investigated via Gene Ontology (GO) analysis (Figure 2D, Appendix A). Interestingly, these genes were found to have a strong association with significant immunomodulatory processes such as the negative regulation of T-cell activation, granulocyte chemotaxis, and the negative regulation of cell adhesion.

To further explore the relationship between GNG4 and immune cell infiltration, the MCP counter, xCell, CIBERSORT, and TIMER algorithms were used to estimate the levels of various types of immune cells from TCGA-BLCA bulk RNA-sequencing data. We showed significantly higher levels of Tregs, CD8^+^T cells, NK cells, CD4^+^T cells, neutrophils, macrophages, and B cells in the high-GNG4 group (Figure 2E,F and Appendix A). Additionally, the high-GNG4 group had a higher level of cytotoxic lymphocytes (Figure 2F). We also used the MCP counter to calculate immune scores and estimate the abundance of various types of immune cells in three validation sets (Appendix A). Similarly, we confirmed substantial immune cell infiltration in the high-GNG4 group in three validation sets. To our surprise, fibroblasts also had a higher valuation score in the high-GNG4 group (Figure 2F). In addition, the number and diversity of T-cell receptors (TCR) were higher in the high-GNG4 group than those in the low-GNG4 group (Appendix A). Taken together, these results also show that the high-GNG4 group had the status of a higher level of immune cell infiltration environment but tended to be exhausted.

Genomic mutation is an important factor in the occurrence of malignant tumors, and this factor affects treatment. Therefore, we employed the ‘maftools’ package to analyze the somatic mutation profile between the high- and low-GNG4 groups in the TCGA-BLCA cohort; the top 10 most commonly mutated genes in each group are shown in Figure 2A. Previously published studies have claimed that genetic variations in the tumor suppressor gene TP53 contribute to human cancers in different ways, whereas TTN mutation was associated with responses to immunotherapy in solid tumors [35,36]. Similar to the previous study, TP53 and TTN were critical genetic alterations in the pathogenesis of BLCA, and these gene alterations were observed in both the high- and the low-GNG4 groups (Appendix A). Notably, RB1, FGFR3, HECTD4, KIAA1109, and TRPM2 occupied the top 5 positions among the different mutant genes between high- and low-GNG4 groups (Appendix A, *p* < 0.001). In recent years, molecular classification (i.e., classification on the basis of genetic alterations and expression) has diversified our knowledge of BLCA and offered a new framework for the stratification and evaluation of responses to different treatment approaches [33,37]. The six consensus classes (LumNS, LumP, LumU, Ba/Sq, stroma-rich, and neuroendocrine (NE)-like) are the most commonly recognized molecular subtypes of BLCA, which can be mainly divided into basal and luminal subtypes. Interestingly, the mutation rate of RB1 was 26.96% in the high-GNG4 group, and RB1 was also mutated in a high proportion in the Ba/Sq group in the molecular classification of BLCA. Similarly, the mutation rate was 21.18% in the low-GNG4 group, which is consistent with the results of mutations in the LumP groups in the molecular classification of BLCA [33]. Thus, related mutations of GNG4 can also directly classify BLCA. Then, we used the MCP counter to estimate the infiltration abundance of different immune cells in GNG4 mutant groups. We defined patients with RB1 and HECTD4 mutations (the related mutated genes in the high-GNG4 group) and those with FGFR3, KIAA1109, and TRPM2 mutations (the related mutated genes in the low-GNG4 group) as the mutation group, and the rest were defined as the wild group (WT). To our surprise, the results showed that the associated mutation group of the high-GNG4 group also exhibited higher immune cell infiltration, whereas the associated mutation group of the low-GNG4 group also exhibited lower immune cell infiltration (Appendix A).

### 3.3. GNG4 as a Biomarker to Predict the Effect of Immunotherapy

As GNG4 is highly expressed in exhausted CD4^+^ T cells and its higher expression is correlated with immune infiltration, we investigated whether GNG4 could be used as a biomarker to predict the immunotherapeutic response to BLCA. Surprisingly, patients with high GNG4 expression had a better response to immunotherapy (Figure 3A). Correspondingly, patients in the high-GNG4 group had better OS and PFS rates than those in the low-GNG4 group (Figure 3B,C). Similarly, results from the IMvogo210 dataset consistently showed that higher GNG4 expression was linked to a better OS rate in patients undergoing immunotherapy with BLCA (Figure 3D). Interestingly, GNG4 predicted the same prognosis for immunotherapy in a melanoma cohort (Figure 3E,F). According to these results, GNG4 may be employed as a biomarker to predict the immunotherapy response in BLCA b, as well as in other cancer types.

### 3.4. GNG4 Is an Indicator of Poor Prognosis and Can Predict the Effect of Chemotherapy

GNG4 is positively correlated with overall survival under immunotherapy, but, we found that higher expression of GNG4 was associated with worse overall survival (OS) and progression-free survival (PFS) rates in BLCA patients who did not receive immunotherapy (Figure 4A,B). Additionally, three BLCA validation cohorts also confirmed that GNG4 expression was an indicator of poor survival in BLCA patients (Figure 4C–E, Appendix A). Meanwhile, subgroup analysis of the pathological stage of tumors in the TCGA-BLCA cohort revealed that the levels of GNG4 expression increased gradually from stage I to stage IV (Figure 4F). Similarly, an identical upward trend of GNG4 expression was also observed in the T-stage subgroup analysis of the TCGA-BLCA cohort (Figure 4G). GNG4 expression was also significantly increased in patients with lymph node metastases and distal metastases (Figure 4G). In particular, we found that GNG4 was highly expressed in metastatic patients. This indicates that GNG4 may also be related to the progress of BLCA. Molecular classifications of BLCA have been divided into six consensus classes (LumNS, LumP, LumU, Ba/Sq, stroma-rich, and neuroendocrine (NE)-like), which are the most commonly recognized molecular subtypes of BLCA [33,37] (Appendix A). Interestingly, we found that the high-level GNG4 expression group was mostly transformed from the basal group (basal/squamous, Ba/Sq) and the stroma-rich group, whereas the low-level GNG4 expression group was mostly from the luminal group (luminal papillary, LumP) (Figure 4H). To examine whether GNG4 could predict the response to chemotherapy in BLCA, we then performed an analysis of the GNG4 gene expression levels in patients with different responses to chemotherapy based on the BLCA data from the TCGA database. The results showed that a better response to chemotherapy occurred in the low-level GNG4 expression group (Figure 4I). Then, we also analyzed the differences in DNA repair gene expression between the high-level and low-level GNG4 expression groups in BLCA. The heatmap shows that the genes of DNA repair were highly expressed in the high-level GNG4 expression group (Figure 4J). Therefore, we concluded that the poor prognosis in the high-level GNG4 expression group might be due to DNA repair, which leads to chemotherapy resistance, whereas the low-level group had a better response to chemotherapy with a better prognosis.

## 4. Discussion

In this study, we found that GNG4 was specifically expressed in a subset of CD4^+^ T cells with exhausted characteristics. The expression of GNG4 is positively correlated with immune cell infiltrations and overall survival under immunotherapy. Interestingly, GNG4 expression is negatively correlated with overall survival and progression-free survival in bladder cancer patients who did not receive immunotherapy, which might be because GNG4 plays a role in developing chemo-resistance in bladder cancer.

GNG4 is an important component of the immune microenvironment in CRC [18]. In our study, we for the first time showed that GNG4 was specifically expressed in CD4^+^ T cells using BLCA single-cell RNA sequencing data. Notably, GNG4 was highly expressed in a cluster of exhausted CD4^+^ T cells, which highly expressed PDCD1, CXCL13, BTLA, and TIGIT. Correspondingly, we found that patients with high GNG4 expression were indeed in an immune-inflamed tumor environment, not immune-excluded or an immune-desert tumor environment [38]. Meanwhile, the levels of effector immune-infiltrating cells, including NK cells, CD8^+^ T cells, macrophages, dendritic cells, and TH1 cells, were also markedly increased in patients with high GNG4 expression. Similarly, the levels of immunosuppressive cells, such as neutrophils, TH2 cells, and Tregs, were also significantly increased. Exhausted cells are transformed from functional T-helper cells or effector cells. Additionally, the high infiltration of exhausted CD4^+^ T cells often causes tumor cells to evade the attack of the immune system, making the overall immune microenvironment present a state of exhausted cells that are enriched but have lost toxic function. However, after immunotherapy, the exhausted state of cells becomes an effector state, which increases the inflammation of the overall microenvironment, which explains the positive association between GNG4 expression and better response to immune checkpoint blockade (ICB) immunotherapy, the first-line treatment for cisplatin-ineligible patients [39]. An effective biomarker to predict ICB immunotherapy response is important for precise treatment. However, although CD8^+^ T effector signatures, hypoxia-related genes, and PVR-related genes have been shown to predict BLCA immunotherapy [40,41,42], these indicators often involve more genes and are difficult to measure. GNG4 as a single gene may have the advantage compared with these biomarkers. Additionally, GNG4 can also predict the immunotherapy of melanoma, which may be an indicator of Pan-cancer. Moreover, GNG4 could also be served as a potential biomarker to predict response to chemotherapy, a common and first-line treatment for BLCA [43]. We found that patients with low expression of GNG4 had a better response to chemotherapy.

Molecular classifications of BLCA have been studied extensively [44,45,46]. The six consensus classes (LumNS, LumP, LumU, Ba/Sq, stroma-rich, and neuroendocrine (NE)-like) are the most commonly recognized molecular subtypes of BLCA, which can be mainly divided into basal and luminal subtypes. We found that patients with high expression of GNG4 are often considered as the basal subtypes with RB1 and TP53 mutations. Patients with low expression of GNG4 are often considered as the luminal subtypes with FGFR3 mutation. This indicates that GNG4 can be used as an important indicator of molecular subtypes, which has not been found in other similar studies [33].

## 5. Conclusions

GNG4 was specifically expressed in exhausted CD4^+^ T cells. Higher GNG4 expression indicates a state of immune inflammation and immune infiltration and can be used as a potential biomarker to predict the response of both immunotherapy and chemotherapy in BLCA.

## Figures and Tables

**Figure 1 genes-13-00693-f001:**
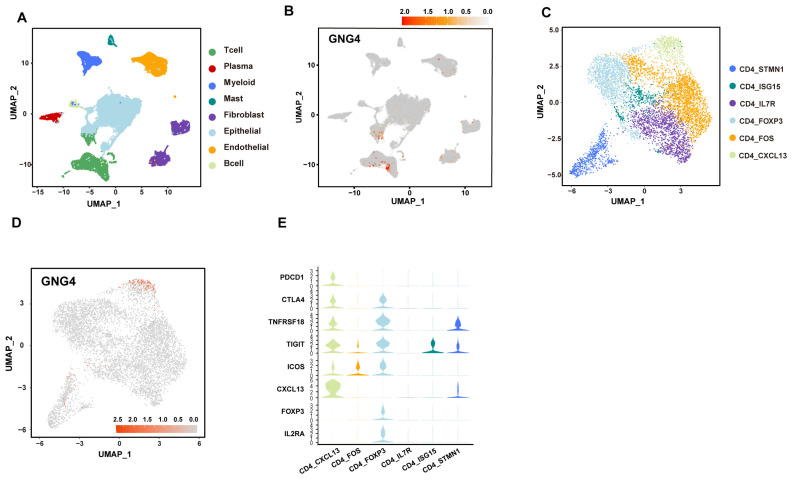
GNG4 is a biomarker for exhausted CD4^+^ T cells population in BLCA. (**A**). The UMAP plot of all cells. Each dot indicates a single cell. Color-coded for the cell type. (**B**). The UMAP plot of GNG4 expression. (**C**). UMAP plot of CD4^+^ T cells. Each dot indicates a single cell, and colors refer to clusters denoted by CD4^+^ T cell types. (**D**). The expression of GNG4 in each type of CD4^+^ T cells. (**E**). Dot plot of immune-related genes expressed in types of CD4^+^ T cells.

**Figure 2 genes-13-00693-f002:**
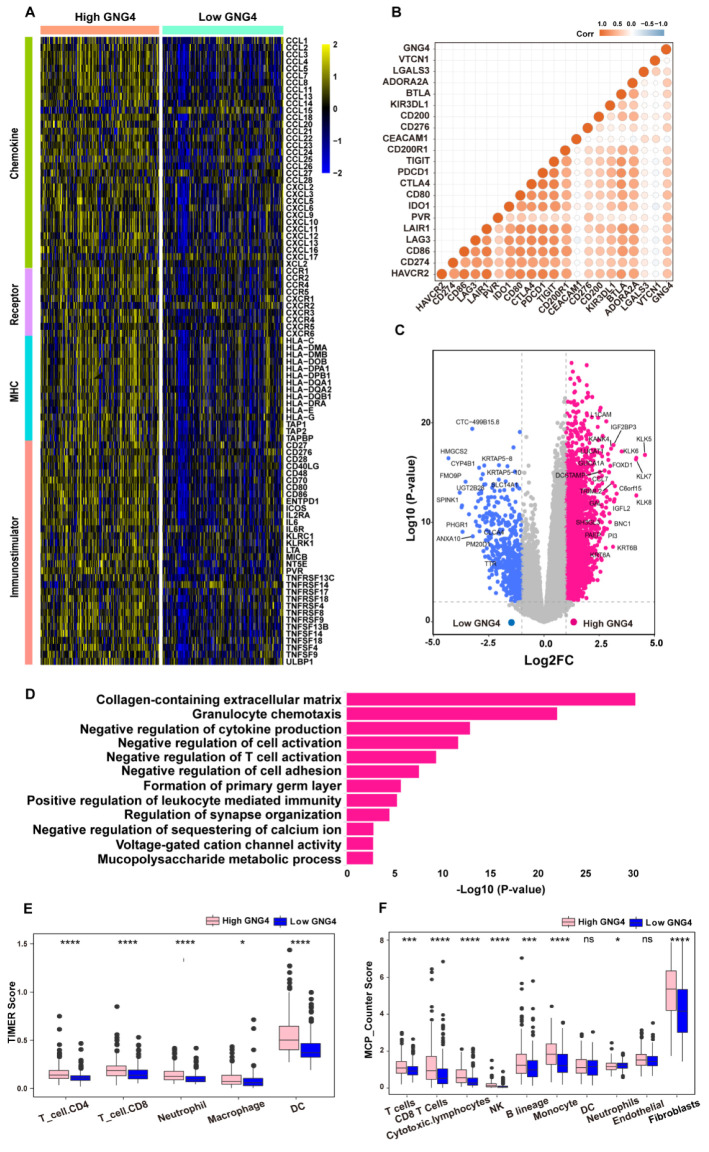
High-GNG4 group exhibits the high expression of immune-related molecules. (**A**). Differential expression of 122 immunomodulators (chemokines, receptors, MHC, and immunostimulators) between high-GNG4 and low-GNG4 groups in BLCA. (**B**). Correlation of GNG4 with 20 inhibitory immune checkpoints. The color indicates the Spearman correlation coefficient. (**C**). Volcano map shows differentially expressed genes between high-GNG4 and low-GNG4 groups in BLCA (high: logFC > 1 and log *p*-value > 2; low: logFC < −1 and log *p*-value > 2). (**D**). Functional comments and pathway enrichment of the genes with GNG4 co-expression genes. (**E**,**F**). Estimation of the abundance of various types of immune cells between the high-GNG4 and low-GNG4 groups using TIMER_counter and MCP_counter algorithm. *, *p*-value ≤ 0.05; *** *p*-value ≤ 0.001; ****, *p*-value ≤ 0.0001; ns, *p*-value > 0.05.

**Figure 3 genes-13-00693-f003:**
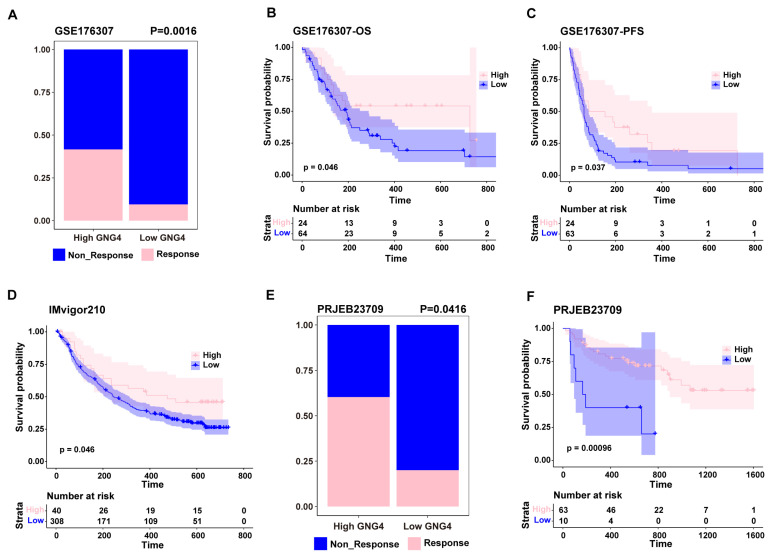
GNG4 as a biomarker to predict the effect of immunotherapy. (**A**). Boxplots showing the proportion of responders between high-GNG4 and low-GNG4 groups in the GSE176307 cohort. Overall survival (**B**) and progression-free survival (**C**) analysis between high-GNG4 and low-GNG4 groups that received immunotherapy in the GSE176307 cohort. (**D**). Overall survival analysis between high-GNG4 and low-GNG4 groups that received immunotherapy in the IMvigor210 cohort. (**E**). Boxplots showing the proportion of responders between high-GNG4 and low-GNG4 groups in the PRJEB23709 cohort (SKCM). (**F**). Overall survival analysis between high-GNG4 and low-GNG4 groups that received immunotherapy in the PRJEB23709 cohort (SKCM).

**Figure 4 genes-13-00693-f004:**
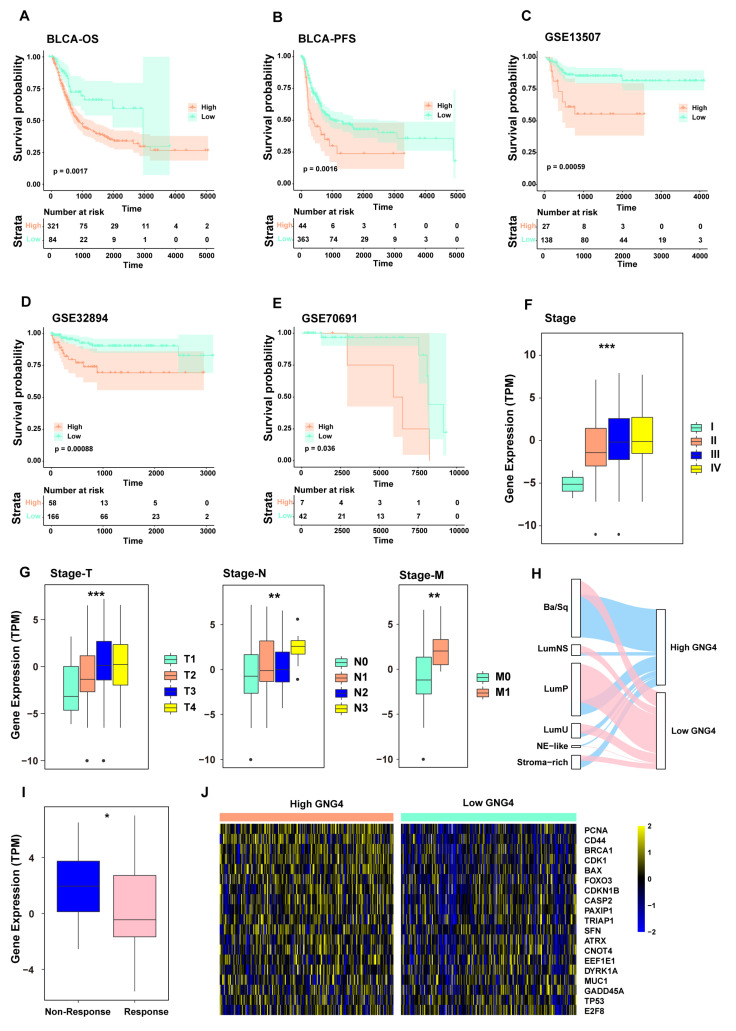
GNG4 as a biomarker to predict survival and chemotherapy effectiveness. (**A**). Overall survival analysis of high and low expression levels of GNG4 in TCGA-BLCA. (**B**). Progression-free survival analysis between high and low expression levels of GNG4 in TCGA-BLCA. (**C**–**E**). Overall survival analysis comparing high and low expression levels of GNG4 in three external, dependent sets: GSE13507, GSE32894, and GSE70691. (**F**). Based on the TCGA data, the expression levels of the GNG4 gene between the different groups of the main pathological stages (stage I, stage II, stage III, and stage IV) of BLCA. *** *p*-value ≤ 0.001. (**G**). Based on the TCGA data, the expression levels of the GNG4 gene between the different groups of the TNM pathological stages (stage T, stage N, and stage M) of BLCA. **, *p*-value ≤ 0.01; *** *p*-value ≤ 0.001. (**H**). Sankey chart displaying the path analysis in patients with BLCA between the published classification systems (LumP, LumU, stroma-rich, LumNS, Ba/Sq, and NE-like) of BLCA and expression levels of GNG4. The line indicates the group; the width of the lines signifies the number of patients who shifted from one state to another (*n* = 406). (**I**). Based on the BLCA data in the TCGA database, the expression levels of the GNG4 gene between the different groups of the response of patients to chemotherapy. *, *p*-value ≤ 0.05. (**J**). Differences in the gene expression of DNA repair between high-level and low-level GNG4 expression groups in BLCA.

## Data Availability

The datasets provided in this study can be found in online repositories. The names and accession number(s) of the repository can be found in the article material.

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
