# Peer review of "G-Protein Subunit Gamma 4 as a Potential Biomarker for Predicting the Response of Chemotherapy and Immunotherapy in Bladder Cancer"

_genes, 2022, doi:10.3390/genes13040693_

Round 1

Reviewer 1 Report

The authors of the article entitled “G Protein Subunit Gamma 4 as A Biomarker for Predicting the Response of Chemotherapy and Immunotherapy in Bladder Cancer” had validated the candidacy of G gamma 4 gene (GNG4) as a biomarker for bladder cancer prognosis using different bioinformatics approaches. The article provides interesting findings about the expression of this gene (GNG4) in bladder cancer and correlated its expression to cancer type, survival, co-expressed inflammatory markers, and therapeutic prognosis. While these findings are of great value in the field and the article would be of great interest for bladder cancer researchers, there are few points that the authors need to consider perfecting the legibility of the text. In general, the text needs some language editing (few examples of grammatical or syntax unclarities are mentioned below) and the discussion/conclusion sections could be revisited to highlight the importance of the findings (some sentences can be moved to the introduction).

Specifically, here are more comments on the text:

  • Title: despite providing very good evidence about the GNG4 in bladder cancer, I would prefer to say “…as a potential biomarker...”, given that more clinical validation is necessary to decide it as a biomarker.
  • Abstract:
    • Line 11-12: “a poor prognostic biomarker.” for what?
    • Line 14: “were” change to “was’
    • Lines 17-18 and 19-20; “Expression of GNG4 can reveal molecular subtypes of bladder cancer.” and “The clinicopathological features could be distinguished by the high and low expression of GNG4.” Aren’t these two statements telling similar findings? It would be important to highlight in the first sentence the OS results regarding GNG4 expression and then specify the BLCA subtype associated with low vs high expression groups.
    • Line 20-21: “GNG4 and GNG4-related……in BLCA” you may specify that this finding refers to the high expression group.
    • Line 24-25: “predictor for immunotherapy” and “predictor for chemotherapy” this was mentioned frequently throughout the article. It would be better to specify what it does predict specifically; response, resistance ...etc.
  • Introduction:
    • Line 46: “therapy” repeated
    • Line 56: “metastasis” is missing between “liver…recurrence”
  • Methods:
    • Line 87: just to mention the source of the single-cell transcriptomic data, was it the same as in 2.1?
    • Check grammar: line 90 change chose/chosen, line 92 change determine/determined, line 94 use/used
    • Line 118-119: p value specification was mentioned twice (in line 115) which one was followed? Two-sided (line 115) or one-tailed (119)? If one-tailed was used for t-test only, would you please explain the reason why this was chosen? Regularly two-tailed is used unless there is a statistical reason for that (considering the direction of the difference between the groups?
  • Results:
    • Line 113-114: GNG4 value as an “early diagnosis of BLCA”. The relative expression levels in early stages as demonstrated in the figure are below zero and it reaches high levels only in advanced stages. I think the statement needs modification.
    • Line 158: (A) in the legend is missing
    • Line 178: like the comment in the abstract: “associated with immunotherapy” it is response or resistance?
    • Line 182: change occupies/occupied, Line 183: interesting/interestingly, Line 205: suggesting/suggested, line 249: indicates/indicating, line 259: delete “also”
    • Line 274: the sentence needs to be rephrased.
  • Discussion:
    • Most of the discussion include restating the results without proper discussion of the results, sometimes the results section had discussion that could be elaborated on here.
    • Line 296: “poor prognosis ..of bladder cancer” some word is missing here, like marker or indicator
    • Line 296: over/overall
    • Line 309-310: I could not understand this sentence, you might kindly rephrase it.
    • Line 320-321: I could not understand this sentence, you might kindly rephrase it.
    • Line 331-332: the authors started a discussion on the chemotherapy, suddenly moved back to immunotherapy. It would be better if the authors give their insight about the goo response to chemotherapy in the low GNG4 group.

Reviewer 2 Report

  1. The Introduction does not explain the work at all. Please write a separate paragraph to explain the rationale of the work, and the overall summary from the analysis.
  2. In a pan-cancer study, how does bladder cancer rank by GNG4 expression?
  3. Are GNG4 high samples enriching for cell proliferation, and/or stemness associated genes?
  4. the author should perform a TIMER and/or CIBERSORT analysis to investigate the level of immune cell infiltration in high, and low GNG4 expressing bladder cancer samples.
  5. Is GNG4 an essential gene for bladder cancer cell survival? make use of DepMap, to investigate the essentiality of GNG4 in bladder cancer.
  6. Databases like GDSC (https://www.cancerrxgene.org/), and others would help to predict the sensitivity of GNG4 high/low expressors to different kinds of drugs. In order to fully utilize information from such bioinformatic studies, the authors should incorporate these analyses in their work as well.
  7. Overall English language of the manuscript is quite poor, with significant grammatical, and typographical errors present throughout the manuscript. The authors must take help from a professional English language proof-editor to edit their manuscript, before re-submission.

Round 2

Reviewer 2 Report

The revised version still lacks a cohesive explanation of the rationale of the study.

Different types of cancer have been compared, without any logical selection criteria.

High expression of GNG4 is associated with more immune cell infiltration, and poor patient outcome. Can the authors comment on the type of cancer cells in BLCA that show high GNG4, from single cell studies? Are these cancer stem like cells, or non-stem cancer cells?

Is the enrichment of inflammation associated pathways only a readout of the cell proliferation status, or is it actually indicative of the malignant quotient of the cells?

In addition, poor English language has made the write-up extremely difficult to navigate through.

Round 3

Reviewer 2 Report

The authors have revised the manuscript to a satisfactory extent. It is suitable for publication now.